# Counteracting Abuse in Health Care: Evaluating a One-Year Drama Intervention with Staff in Sweden

**DOI:** 10.3390/ijerph17165931

**Published:** 2020-08-15

**Authors:** Anke Zbikowski, A. Jelmer Brüggemann, Barbro Wijma, Katarina Swahnberg

**Affiliations:** 1Women’s Clinic, Ryhov County Hospital, 55185 Jönköping, Sweden; anke.zbikowski@rjl.se; 2Department of Thematic Studies—Technology and Social Change, Faculty of Arts and Sciences, Linköping University, 58183 Linköping, Sweden; jelmer.bruggemann@liu.se; 3Department of Biomedical and Clinical Sciences (BKV), Faculty of Medicine and Health Sciences, Division of Children’s and Women’s Health (BKH), Linköping University, 58183 Linköping, Sweden; barbro.wijma@liu.se; 4Department of Health and Caring Sciences, Faculty of Health and Life Sciences, Linnaeus University, Hus Vita, 39182 Kalmar, Sweden

**Keywords:** abuse in health care, Forum theater, role-play, ethical learning, professional-patient relation, communication training

## Abstract

In Northern European countries 13–28% of female patients seeking gynecological health care have reported abuse by health care staff (AHC). We conducted workshops with health care staff using the improvised role-play method Forum Play (FP), based on techniques developed by Boal. The study explores to what extent the intervention increased the staff’s awareness of AHC and their ability to take action against it. A total of 16 half-day FP workshops were conducted with staff from a Swedish women’s clinic over one year. Self-reported questionnaires were distributed to all staff before, during, and after the intervention. Primary outcome measures were the number of reported occasions of AHC and FP participants’ ability to act in AHC-situations. We found an increase in the participants’ self-reported ability to act in AHC-related situations. However, no change could be observed in the number of reported occasions of AHC between baseline and one year after the intervention. Health care staff’s participation in workshops using improvised role-play can increase staff’s perceived ability to take action in AHC situations. The voluntary nature of the intervention may have attracted those who were already aware of the topic, and likely explains the unchanged awareness of AHC.

## 1. Introduction

It is not uncommon that patients feel abused by health care staff. Recent reports of disrespect and abuse during faculty-based maternity care have gained global recognition [1,2,3,4,5,6,7]. In a cross-sectional study with 3641 women in five Northern European countries using the NorVold Abuse Questionnaire (NorAQ) it was estimated that 13–28% of female patients seeking any kind of gynecological health care had experienced abuse in health care (AHC) in their lifetime [8]. The aforementioned study defined AHC as patients’ experiences in health care settings according to the operationalization in Table 1. Emphasizing the subjectivity of experiences in the current study, AHC is not confined to events demanding legal prosecution.

A common characteristic of AHC is the experience of dehumanization [9,10,11], or a situation “…devoid of care, where patients suffer and feel they lose their value as human beings” [12] (p. 123). Concerning maternity care, AHC experiences are associated with fear of childbirth and postpartum depression [13,14]. As a possible consequence patients may avoid contacting the health care system which can lead to the deterioration of patients’ health, prolonged suffering, increased treatment expenses, and low return to work rates [9,15]. In contrast to measurements to protect health care staff from being exposed to physical or verbal abuse at their workplace [16,17], little is known about effective strategies to prevent patient abuse by staff. From an ethical point of view this is highly problematic. Concerning the prevention of AHC, the practical value of ethical guidelines for health care professionals seems to be limited [18]. However, a study by Swahnberg et al. indicates that it is not the staff’s lack of theoretical knowledge that allows AHC to happen, but rather the appraisal of AHC as “ethical lapses” [19]. Moreover, the awareness of AHC seems to vary according to the context and the staff members’ possibility to act in a given situation [20]. A lack of experience, being afraid of challenging their seniors, and fear of negative consequences for their career have been described as reasons for not responding to AHC in a study with South African medical students [21]. Thus, strategies aimed at reducing the occurrence of AHC ought to have two objectives: promoting an increased awareness and understanding of situations that are experienced as AHC and enabling staff to recognize more possibilities to take action and to develop a broader range of courses of action.

Research has shown that inhumane behavior and violence becomes possible when remaining unnoticed, ignored, tolerated, accepted, or reinterpreted [22,23,24]. Glover describes how the development of violence is linked to an erosion of an individual’s moral resources, i.e., human responses such as respect and sympathy, and a person’s moral identity. He points out a number of processes that effectuate this erosion, such as passivity, the ’habit of participation’, fragmentation of responsibility, deficit of moral imagination, degradation of others, and pressure to conform or obey [22]. We assume that these mechanisms apply to AHC as well. By providing a platform where situations of AHC are identified, explored, and acted upon, reflection and reevaluation of one’s own perceptions and actions should become possible. If AHC is acknowledged and discussed by staff, its impact on the climate at a clinic could be expected to lead to even larger awareness of AHC.

Following on from these assumptions, in the current study, a drama intervention was used to counteract AHC. Activation of spectators is one of the central ideas in the interactive theater method of Forum Play (FP), a Swedish modification [25,26] of Forum Theater developed by Boal [27]. In FP one group of participants improvise a situation of oppression or power imbalance in a role-play, and the other participants constituting the audience are encouraged to engage and try out ways of acting to counteract the oppression. The aspect of working with improvised role-play in a defined group distinguishes FP from Forum Theater. Furthermore, in FP, value exercises help participants to reflect on their motives and attitudes [26]. FP provides both a rational approach facilitating intellectual understanding, and an experiential aspect by offering possibilities to test different variations of intervening by acting them out. For this project, FP was therefore considered an appropriate method to enhance staff’s awareness of AHC and increase their ability to counteract AHC. In the current article we report on an evaluation of a one-year intervention with FP workshops at a Swedish clinic.

When examining staff’s attitudes and motives before and after the intervention we assumed that we would find that the everyday conversations between those participating in FP (FP participants) and those who were not participating (non-participants) would lead to a greater openness to making AHC the subject of a discussion at the clinic, thus facilitating an increased awareness of situations of AHC (Hypothesis 1). Accordingly, the number of instances of AHC reported by all respondents would increase during and after the workshop series and would remain unchanged or decrease (wear-off effect) after one more year (Hypothesis 1a). The effect was expected to be more explicit among FP participants than in the total sample (Hypothesis 1b). This hypothesis is based on the results of two qualitative studies. These indicate that health care staff do not recognize AHC as such, hence prior to FP it does not receive their attention [19]; and staff’s awareness varies according to context and possibilities to act, hence the participants’ awareness should show a more marked increase after FP [20].

We also expected an increased ability to counteract AHC or to minimize patients’ suffering from AHC. FP participants were expected to become able to recognize more possibilities to act in situations with a risk of AHC, and to act more according to their own moral beliefs (Hypothesis 2). This hypothesis is based on Glover’s theoretical elaboration on the development of violence which becomes possible through a gradual erosion of moral resources [22]. Strengthening these resources is expected to be an important factor in enabling staff to counteract AHC, especially when involved staff members feel obliged to act in order to help the patient in a potentially abusive situation, while at the same time perceiving conflicting moral obligations, such as not wronging others involved in the situation (moral dilemma).

## 2. Materials and Methods

A women’s clinic at a county hospital in Sweden agreed to invite all staff members (average number per year during the study period n = 133) to voluntarily participate in one or more FP workshops addressing the issue of AHC over a period of 13 months. A total of 16 FP workshops that each lasted 3–3.5 h were organized and conducted by a professional FP leader, who worked closely with the research team but is not a co-author of this article. Prior to the project, three information meetings were organized for the entire staff. The staff received written and oral information about the aim and the content of the workshops. Information about the project was accessible through the clinic’s intranet, which also provided the possibility to register for the workshops, which took place during working hours. Of 136 staff members who had the possibility to participate, 76 (56%) gave their informed consent and participated in at least one workshop. Twenty-six staff members or 34% of the FP participants took part in two or more workshops.

For reasons of workability the number of participants per workshop was limited to 15 and workshops with less than four participants were canceled. During the workshop period, between one and three workshops were conducted every month with exceptions for December and for the summer months when hospital capacities in Sweden are reduced to a minimum because of vacations. Providing workshops over a period of 13 months allowed the clinic’s organization to adapt working schedules so that all staff had the possibility to participate in at least one workshop. Still, organizational conditions had an impact on both the number of participants per workshop and the composition of the groups. For most of the physicians, participation was not possible until a workshop was conducted as part of their regular internal education. All of the other workshops were relatively inhomogeneous concerning profession and age distribution, though with an overrepresentation of female midwives, who constitute the largest group of professionals at the clinic (Table 2).

All participants of the workshops were asked to share the story of an AHC situation they had experienced or heard about. Then the group chose two to three of the situations they wanted to work with. The group then divided up in subgroups and each subgroup created a short scene of one of the stories chosen by the subgroup. When one subgroup was playing their scene, all the other participants watched the scene as active spectators, also called “spect-actors” by Boal [28]. A typical scene could be a doctor examining a woman in labor in a way that caused the patient great pain, ignoring the woman’s desperate plea to stop. Assuming that AHC is happening because it is not counteracted by those who are witnessing the abusive action, the spectators were encouraged to think about alternative ways of acting for one of the witnesses or bystanders in the scene. Then the scene was replayed. The spectators were asked to stop the scene as soon as the abuse happened or was about to happen, to go “on stage” and to exchange one of the bystanders or witnesses in the scene. In the case of our example it could have been a midwife or an auxiliary nurse. The scene was now replayed with the replaced witness who could try to counteract the abusive behavior not only by using language, but also movements and body language. For example, by replacing a midwife in the scene the spectator could try to stop the doctor by placing a hand on the doctor’s arm. Afterwards the effect of the intervention was discussed by actors and spectators. Every spectator was invited to try one or more possibilities to prevent the abuse or to alleviate its consequences. The procedure of replaying the scene, stopping it, and replacing one of the actors was repeated until the group found at least one workable solution. The interventions of the spectators and the final discussions were moderated by the FP leader, or in Boal’s terminology the “joker” [28].

In each workshop, new stories were recollected by the participants and even if some stories were similar to each other, the new groups always generated new solutions. Thus, the repeated participation could help to broaden the participant’s repertoire of actions in AHC situations and was therefore considered beneficial.

For the evaluation of the intervention the authors developed a questionnaire. It contained questions about background characteristics such as age, sex, and profession. Moreover, it contained the following questions addressed to all staff about the frequency of observed incidents of AHC (Hypothesis 1) and the perceived impact of AHC: “Have you heard any stories about the abuse of a patient in health care? How many times?” as well as questions addressed to the FP participants about the experience of FP and the perceived impact of FP (Hypothesis 2): “Have you been involved in a situation where a patient has been abused in health care? How many times have you been involved?“ In the questionnaire, AHC was defined as a failed encounter in health care, implying that a patient experienced being hurt or humiliated. Only those questions that are operationalized for exploring the hypotheses are included in this article. Due to a risk of selection bias in workshop participation we will not compare FP participants with non-participants in this study.

The questionnaires (Q) were distributed before (QI, t₀) during (QII, t₀ + 5 months), directly after (QIII, t₀ + 14 months), and one year after the workshop series (QIV, t₀ + 25 months). For QII, QIII, and QIV we used a shortened version of QI. All questionnaires were sent to the homes of all staff members with a stamped and self-addressed envelope. Two reminders were sent to non-respondents. Returned questionnaires were anonymized and encoded.

All personnel who were employed on a regular basis at the host clinic from the time of the distribution of the first questionnaire until the distribution of the last one (t₀–t₀ + 25 months) were eligible for the evaluation, irrespective of workshop participation (Figure 1). The background characteristics of all staff at the target clinic and of staff that participated in FP are shown in Table 2.

The mean value, standard deviation, and median were calculated for discrete values. Proportions in percentage of valid answers, and the median and quartiles were used to describe ordinal variables. At least two completed questionnaires were needed for the testing of Hypothesis 1, and for the testing of Hypothesis 2, cases were selected where respondents had reported participation in FP workshops in at least two completed questionnaires. For the testing of Hypothesis 1, time intervals of similar size (t₀, t₀ + 14 months and t₀ + 25 months) were chosen. The number of reported incidents of AHC for one year was assessed by adding the counts of AHC incidents reported by staff in QII to those in QIII.

Both hypotheses were tested for no change by applying the Wilcoxon signed rank test for assessing the pattern of change of the ordered categorical data and McNemar’s test for dichotomous categorical data.

The study was approved by the ethical review board in Linköping, Sweden (reg. no. 194–06).

## 3. Results

The response rates were: 70% for QI (92/131); 63% for QII (85/134); 59% for QIII (79/133), and 59% for QIV (78/132) (Figure 1). According to hypothesis 1a in the total sample, the pattern of change was expected to be characterized by an initial increase of the number of reported AHC incidents in QIII that would then remain unchanged or decrease in QIV (Table 3, questions 1–4). This pattern of change was expected to be more explicit in the group of FP participants according to hypothesis 1b. However no change could be shown for the number of occasions where staff had heard stories of AHC (Table 3, questions 1 and 2) or had been involved in AHC (Table 3, questions 3 and 4) in the sample that included all respondents, nor in the subsample of FP participants. Due to the lack of significant results, the results for questions 1–4 for the total sample are not displayed in detail. The results for FP participants are displayed in Table 3 for reasons of coherence.

According to Hypothesis 2, FP participants were expected to report an increase in their ability to act when caught in the moral dilemma of an AHC situation. Their way of handling such a situation was expected to correspond more often to what they felt was the right thing to do (Table 4, questions 5–8). In all three questionnaires, an increase of the perceived ability to act in situations with a moral dilemma was reported (Table 4, question 5). The median of estimated increase was higher after the workshop series in QIII, than during the workshop series in QII. The increase of the perceived ability to act in situations with a moral dilemma was estimated to be significantly higher after the workshop period in QIII and QIV than at the beginning of the workshop period QII.

The number of reported occasions where FP participants acted more according to their own moral beliefs than before the intervention (Table 4, question 7) was increased during the workshop series in QII and was even higher after the workshop series in QIII. Due to the construction of the questionnaire—in QII and QIII question 8 was only answered by those who had answered question 7, while QIV only contained question 8 and not 7, in order to shorten QIV—the number of valid answers included in hypothesis testing in question 8 was low. The degree to which handling an AHC situation involving a moral dilemma corresponded to what participants felt was the right thing to do was estimated as moderate in all questionnaires (Table 4, question 8).

The answers of the two variables in QIV that were used to measure awareness (questions 1 and 3, Table 3) show a moderate correlation (Spearman’s rho 0.58, *p* = 0.01). A high correlation of the answers to the question 5 (Table 4) with answers to the question 7 (Table 4) in QIV was found (Spearman’s rho 0.84, *p* = 0.01) supporting the internal consistency of these variables.

## 4. Discussion

The results of the present study indicate that participation in FP based on an educational workshop program increases the self-reported ability to act more according to what staff perceive to be morally adequate actions in situations they feel risked being experienced as abusive by patients. One year after the end of the workshop series, staff members who had participated in FP still reported an increase in their perceived ability to act. Additionally, the participants in FP workshops had experienced more often that they were able to act according to their own moral beliefs. Our results are consistent with research confirming that role-playing is an effective way to enhance empathy and to improve the communication skills of health professionals [29,30]. They also support the findings that showed that participants in FP workshops had experienced a change in their motivation and perceived ability to act in situations of AHC [31,32,33]. However, while these studies gave reason to assume that FP workshop training increased the awareness of AHC as a shared problem, such a change could not be seen in the quantitative evaluation presented in this article.

The study’s strength is the uniqueness of its quantitative long-term follow-up of an intervention with FP for health care staff over two years. An intervention with FP addressing AHC has not been applied and evaluated in this context and to this extent before. The response rates for any combination of two questionnaires were good, which was necessary for the assessment of change. In contrast to the qualitative studies assessing the impact of FP interventions on health care staff having participated in FP workshops with a focus on AHC [31,32], the questionnaires in this study reached all staff members at the clinic. The anonymity of the questionnaires made it more probable that staff answered honestly. In a culture of silence around AHC, guilt and shame may prevent staff from disclosing their experiences when openly confronted with the issue of AHC [34].

However self-assessment questionnaires are prone to errors resulting from recall difficulties and misunderstandings. The experimental character of this study implies a lack of evaluated measurement methods and may entail weaknesses in the design of the questionnaires. The operationalization of variables to assess openness and awareness for AHC at the clinic was not optimal. For reasons of confidentiality, talking with non-participants about examples from FP based on real cases was discouraged. Thus, the conditions for AHC becoming a topic of everyday discussion were unfavorable. Furthermore, not all of the clinic’s staff participated in the FP workshops. Compared to the proportions of the different professions at the clinic, fewer auxiliary nurses and secretaries were represented in the group of participants (Table 2). However, as auxiliary nurses often have the task to assist physicians and midwives, they are particularly likely to be in a bystander position. In the FP workshops the position of the witness or bystander was the position from which acting was considered to be possible and therefore actors being in this position were replaced by the spectators. So auxiliary nurses may be the professional group that would have had the most benefit from FP. Moreover, as the informational meetings showed, the project was not uncontroversial. An influence of critical voices among staff on the atmosphere of the clinic can be assumed. The inclusion of all staff members in FP workshops, even those who had reservations about participating or a lower interest in FP, could possibly have generated other results. For example, a study on a Forum Theater approach for university students to promote engagement in bystander interventions to prevent rape, showed substantial increases in the perceived likelihood of engaging in bystander interventions in those with lower expectations about the effectiveness of bystander interventions [35].

Due to the voluntary nature of the intervention it can be considered that staff members who had participated in FP also had an interest in the issue of AHC. According to this assumption, participation in FP would not necessarily result in an increase of awareness but only in an increase in the participants’ ability to act. In a qualitative evaluation Swahnberg and Wijma showed an increased sense of urgency to act against AHC among those informants who had participated in FP workshops [31,32], which is reconcilable with the results of this study. The main results of the present study are in line with the above-mentioned qualitative studies, which support the validity of the ability questions in our questionnaires. Still, the voluntariness of the participation in FP carries the risk of a selection bias. Therefore, future studies are needed firstly, to find additional ways to assess a change in awareness and the ability to act, and secondly, to investigate if FP is workable and beneficial in a mandatory setting.

The experiences and the findings of this study may contribute to the development of more refined instruments to assess the effects of FP. Nevertheless, it is difficult using quantitative methods to measure improvement of awareness and inter-relational abilities. Assessing the degree of an internal struggle or the success of moral reasoning and acting in a situation involving a moral dilemma is limited when using questionnaires. Research in related areas aimed at assessing changes of elusive properties such as e.g., moral distress [36] shows that it remains difficult to distinguish between a lack of a measurable change and a possible inadequacy of the assessment method.

For the development of strategies to prevent AHC, the current study contributes in two ways. Firstly, it supports earlier reports that FP has the potential to empower participants and to widen their repertoire and their willingness to act in situations involving a moral dilemma [37]. At the same time, the difficulties in this study point at a risk that the effect will be limited to a selected group. To avoid ambiguity in the measurement results, FP should be implemented in a way that includes the entire staff. In this way, AHC could be a shared issue, and skills for preventing AHC could be improved and assessed on a clinic level. Secondly, the need to improve quantitative methods for measuring FP’s effectiveness regarding the prevention of AHC is highlighted. This also shows that findings from qualitative research are not as easily captured quantitatively. Behavioral change is a complex issue and difficult to assess. Individual, social, and situational factors are involved and our knowledge about their impact is limited. Abusive behavior must be prevented, but strategies for prevention have to take into account that shame and guilt may lead to the persistence of AHC. Creating an atmosphere of openness and awareness seems the most difficult and important part of the process. Refining research on FP as a method to prevent AHC and integrating the knowledge from qualitative and quantitative studies is all the more necessary in order to be able to compare FP to other possible interventions.

## 5. Conclusions

The results of this two-year follow-up study show that staff members who have participated in theater workshops report that they perceive and experience an increased ability to act according to their moral beliefs in AHC situations. The study, however, could not confirm an increased awareness of AHC. These results should be cautiously interpreted with decreasing response rates and voluntary participation in workshops in mind. Future applications of FP as a strategy to counteract AHC should consider an approach that includes all staff members and not a selected group. Moreover, the need for refining evaluation methods is emphasized.

## Figures and Tables

**Figure 1 ijerph-17-05931-f001:**
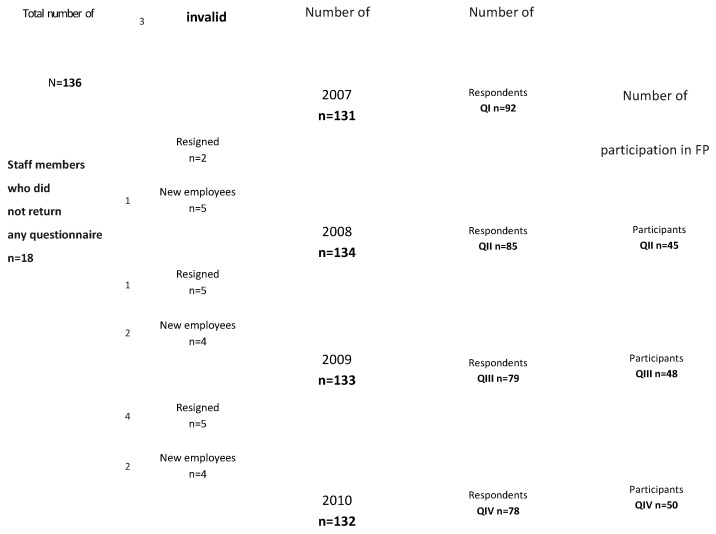
Number of employees who completed the questionnaires and participated in Forum Play (FP).

**Table 1 ijerph-17-05931-t001:** Questions about AHC in NorAQ [8].

**Mild abuse**	Have you ever felt offended or grossly degraded while visiting health services, felt that someone exercised blackmail against you, or did not show respect for your opinion—in such a way that you were later disturbed by or suffered from the experience?
**Moderate abuse**	Have you ever experienced that a ‘normal’ event, while visiting health services suddenly became a really terrible and insulting experience, without you fully knowing how this could happen?
**Severe abuse**	Have you experienced anybody in health service purposely—as you understood—hurting you physically or mentally, grossly violating you or using your body and your subordinated position to your disadvantage for his/her own purpose?

Answer alternatives (the same for all questions) 1 = No; 2 = Yes, as a child (<18 years); 3 = Yes, as an adult; 4 = Yes, as a child and as an adult (AHC is assumed in cases where the answers 2, 3, or 4 were given).

**Table 2 ijerph-17-05931-t002:** Background characteristics of staff at the host clinic.

	All Eligible Staff at the Host Clinic ^1^	Participants in Forum Play (FP) ^2^	
Total	136	76	(56%)
Physicians	20	13	(65%)
Midwives	76	47	(62%)
Auxiliary nurses	29	11	(38%)
Secretaries	11	5	(45%)
Female	126	71	(56%)
Male	10	5	(50%)
Age median	49.9	49 years ^3^	
Age range	20–>65	28–63 ^3^	

^1^ Clinic data, ^2^ data from the workshop registration, ^3^ data from the FP participants returning at least one questionnaire (n = 61).

**Table 3 ijerph-17-05931-t003:** Variables relevant for Hypothesis 1: FP participants’ awareness of AHC.

		QI (t₀)	QIII (t₀ + 14 Months)	QIV (t₀ + 25 Months)	Change QI-QIII	Change QIII-QIV	Change QI-QIV
1. Have you heard any stories about the abuse of a patient in health care? ^2^	n	48	49	49	39	42	38
	No	20 (42%)	30 (61%)	20 (41%)			
	Yes, once	15 (31%)	9 (18%)	19 (39%)			
	Yes, several times	13 (27%)	10 (20%)	10 (20%)			
	Mc Nemar’s test ^1^				*p* = 0.077	*p* = 0.077	*p* = 0.804
2. How many times [have you heard about AHC]? ^2^	n	41	54	43			
	Mean	1	0.89	0.74			
	Standard deviation	1.82	1.59	0.98			
	Wilcoxon signed rank test				n.a.	n.a.	n.a.
3. Have you been involved in a situation where a patient has been abused in health care? ^2^	n	48	47	51	40	41	40
	No	41 (85%)	40 (85%)	35 (69%)			
	Yes, once.	2 (4%)	4 (9%)	13 (25%)			
	Yes, several times	5 (10%)	3 (6%)	3 (6%)			
	Mc Nemar’s test ^1^				*p* = 1.000	*p* = 0.920	*p* = 0.065
4. How many times have you been Involved [in a situation where a patient has been abused in health care]? ^2^	n	45	55	49			
	Mean	0.18	0.24	0.35			
	Standard deviation	0.65	0.82	0.69			
	Wilcoxon signed rank test				n.a.	n.a.	n.a.

**Legend:** QI-QIV = questionnaire 1–4; n.a.= not applicable. **Notes:**
^1^ Dichotomized question (yes/no), ^2^ Variations: In QI the question relates to last 12 months, in QII—QIV the question relates to the time since the last questionnaire.

**Table 4 ijerph-17-05931-t004:** Variables relevant for Hypothesis 2: FP participants’ ability to act according to their moral beliefs in situations with a moral dilemma.

		QII (t₀ + 5 Months)		QIII (t₀ + 14 Months)		QIV (t₀ + 24 Months)		Change QII-QIII	Change QIII-QIV	Change QII-QIV
5. Do you think that Forum Play has affected your ability to act in a way that feels right to you in situations involving a moral dilemma in health care? Your ability has increased ^1,2^	n	45		48		51		32	38	35
	Median (quartile 1|quartile 3)	3.0	(2.0|6.0)	5.0	(3.0|6.0)	5.0	(3.0|6.0)			
	Wilcoxon signed rank test							***p* = 0.039**	*p* = 0.127	***p* = 0.012**
6. Do you think that Forum Play has affected your ability to act in a way that feels right to you in situations involving a moral dilemma in health care? Your ability has decreased ^1,2^	n	45		48		51				
	Median (quartile 1|quartile 3)	0	(0|0.5)	0	(0.0|2.0)	0	(0|1.0)			
	Wilcoxon signed rank test							n.a.	n.a.	n.a.
7. After your participation in Forum Play, did your way of handling a health care situation involving a moral dilemma correspond more to what you felt was the right thing to do?	n	43		43				10		
	No	10	23%	8	19%					
	No, I have not experienced such situations ^3^	26	61%	20	47%					
	Yes, once.	6	14%	4	9%					
	Yes, several times	1	2%	11	26%					
	Wilcoxon signed rank test							***p* = 0.023**	n.a. ^4^	n.a. ^4^
8. After your participation in Forum Play, to what degree did your way of handling a health care situation involving a moral dilemma correspond to what you felt was the right thing to do? ^2^	n	6		14		48		4	14	6
	Median (quartile 1|quartile 3)	5.50	(2|7.0)	6.0	(4.75|7.3)	5.0	(3.0|6.0)			
	Wilcoxon signed rank test							*p* = 1.000	*p* = 0.605	*p* = 0.785

Legend: QI-QIV = questionnaire 1–4; n.a.= not applicable. Note: ^1^ Variations: In QI the question relates to last 12 months, in QII–QIV the question relates to the time since the last questionnaire. ^2^ Answering alternative VAS (visual analogue scale) 0–10 (min-max). ^3^ Answering alternative not included in hypothesis testing. ^4^ Question 7 was not asked in questionnaire IV.

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
