# Peer review of "Counteracting Abuse in Health Care: Evaluating a One-Year Drama Intervention with Staff in Sweden"

_ijerph, 2020, doi:10.3390/ijerph17165931_

Round 1
Reviewer 1 Report
The authors need to provide a more detailed discussion of the Forum Play that was used.
for example, how many people were part of the forum? What was the mix, e.g., age, sex, of the forum.
Other than my suggestion above, the research is well presented, carried out and analyzed. I liked the discussion section as the authors acknowledge the shortcomings of the research. Nevertheless, they will build on this foundation.
Reviewer 2 Report
The strenght of this study is its prospective and follow-up nature. The area covered in the publication is very important and has a real impact on the quality of care for a woman. The selection of participants was described well.
The limitations of the study were properly described. It is worth supplementing the method section with an example of the topic of the scene, the action played by the participants. I propose to add information whether the FP leader was one of the researchers. It would be useful to add "seniority" (if checked) to the characteristics of the studied group.
1. Was the professional FP leader a co-author of this study?
2. Was the work experience of the participants surveyed?
2. It would be worth giving an example of the topic of the scene, the action played by the participants.
Reviewer 3 Report
This manuscript is well written and addresses a critical and fundamental issue in abuse of patients in health care. Indeed, research has focused on the experiences of patients but none of the studies had focused on the role of the by stander or other staff members involved in the care of patients while the abuse is happening.
- I have a few comments that the authors should address to improve clarity on the methods. For example, why 16 workshops? Was the context or content of the workshops similar or different? How many workshops were attended by individual participants? What were the benefits of attending more than one workshops?
- Could the authors state that some of the questions could only be answered by the FP participants since the evaluation included all personnel who were employed on a regular basis at the host clinic.
- Although limitations are identified and implications extensively discussed, the risk of selection bias in workshop participation is only mentioned in the introduction but not clarified under study limitations.
